# Pathway of Hsp70 interactions at the ribosome

Kanghyun Lee[1,4,6], Thomas Ziegelhoffer[1,6], Wojciech Delewski[1], Scott E. Berger [1,2,5], Grzegorz Sabat[3] & Elizabeth A. Craig [1✉]

In eukaryotes, an Hsp70 molecular chaperone triad assists folding of nascent chains emerging from the ribosome tunnel. In fungi, the triad consists of canonical Hsp70 Ssb, atypical Hsp70 Ssz1 and J-domain protein cochaperone Zuo1. Zuo1 binds the ribosome at the tunnel exit. Zuo1 also binds Ssz1, tethering it to the ribosome, while its J-domain stimulates Ssb's ATPase activity to drive efficient nascent chain interaction. But the function of Ssz1 and how Ssb engages at the ribosome are not well understood. Employing in vivo site-specific crosslinking, we found that Ssb(ATP) heterodimerizes with Ssz1. Ssb, in a manner consistent with the ADP conformation, also crosslinks to ribosomal proteins across the tunnel exit from Zuo1. These two modes of Hsp70 Ssb interaction at the ribosome suggest a functionally efficient interaction pathway: first, Ssb(ATP) with Ssz1, allowing optimal J-domain and nascent chain engagement; then, after ATP hydrolysis, Ssb(ADP) directly with the ribosome.

[1] Department of Biochemistry, University of Wisconsin-Madison, Madison, Wisconsin 53706, USA. [2] Department of Chemistry, Lafayette College, Easton, PA 18042, USA. [3] Biotechnology Center, University of Wisconsin-Madison, Madison, Wisconsin 53706, USA. [4]Present address: Department of Biochemistry and Biophysics, Institute for Neurodegenerative Diseases, University of California, San Francisco, CA 94158, USA. [5]Present address: Biophysics Program, Stanford University, Stanford, CA 94305, USA. [6]These authors contributed equally: Kanghyun Lee, Thomas Ziegelhoffer. ✉email: ecraig@wisc.edu

Folding and trafficking of nascent polypeptide chains as they exit the ribosome are inherently problematic, as thousands of polypeptide chains with differing sequences need to be accommodated[1,2]. The area surrounding the exit of the ribosome tunnel through which polypeptides pass is an interaction hub for proteins that facilitate protein folding and/or targeting[3–7]. The focus of this report is the ribosome-associated Hsp70 molecular chaperone system important for nascent chain folding in eukaryotes[3,8]. As in all Hsp70 systems, a J-domain protein is a key component—acting, via its J domain, as an Hsp70 cochaperone to drive the hydrolysis of Hsp70-bound ATP, promoting trapping of substrate polypeptides[9,10]. The J-domain protein of this system[11–13] is generically called Zuotin, more specifically Zuo1/DNAJC2 in fungal/human cells, respectively. In most eukaryotes, the abundant, soluble cytosolic Hsp70 partners with Zuotin, along with numerous other J-domain proteins. However, fungi have, in addition to the soluble Hsp70 Ssa, a specialized ribosome-associated Hsp70 called Ssb (encoded by SSB1 and SSB2 in Saccharomyces cerevisiae)[14], which partners with Zuo1. Unlike other J-domain proteins, Zuotin forms a stable heterodimer with an "atypical" Hsp70, called Ssz1 in fungi and HspA14 in human cells[15,16]. Zuo1 also interacts directly with the 60 S subunit, thereby tethering Ssz1 to the ribosome. This complex (often called the ribosome-associated complex, RAC) is approximately one-third the abundance of ribosomes[17]—with virtually all ribosome-associated[18–21]. The absence of any one of the yeast triad components (Ssb, Ssz1, or Zuo1) results in pleiotropic phenotypes, including slow growth, particularly at low temperatures, and sensitivity to cations[15,22,23].

Though specialized, in that a high proportion is found associated with ribosomes, Ssbs are canonical Hsp70s—they have the typical Hsp70-domain architecture and undergo efficient, adenine nucleotide-dependent cycles of interaction with substrate polypeptides[24,25]. ATP hydrolysis is key to function, as it drives the necessary large-scale conformational changes of Hsp70 domains—the N-terminal nucleotide-binding domain (NBD) and the C-terminal substrate-binding domain (SBD) (Fig. 1a). In the ATP-bound state, the SBD is docked on the NBD, making the substrate-binding site in the β-subdomain of the SBD (SBDβ) easily accessible. Upon ATP hydrolysis, conformational changes result in domain disengagement and trapping of the substrate by "clamping" of the α-subdomain (SBDα) over the peptide-binding cleft of SBDβ. Timing of ATP hydrolysis is regulated by J-domain interaction at the NBD–SBDβ interface and substrate interaction in the SBDβ binding cleft. The cycle is completed by the action of a nucleotide-exchange factor (NEF), facilitating release of ADP, that is followed by binding of ATP and substrate release[10,26].

Ssz1 is called an atypical Hsp70 because it does not undergo the efficient, ATP-regulated substrate-interaction cycle that characterizes canonical Hsp70s. Although Ssz1 binds ATP, little or no hydrolysis occurs, and thus is maintained in the ATP-bound conformation[21,27,28]. Furthermore, though the divergent C terminus forms a SBDβ-like subdomain, Ssz1 is truncated, having no SBDα. The unusual stability of the Ssz1–Zuo1 complex is due to intertwining of a segment near Zuo1's N-terminus at the Ssz1 NBD–SBD interface[29,30]. Ssz1 is not the only atypical Hsp70 in the eukaryotic cytosol. Hsp70 Sse also binds ATP, but has marginal ATPase activity. It acts as one of the NEFs for canonical Hsp70, forming, via its NBD and SBDα, transient heterodimers with the NBD of both Ssb(ADP) and Ssa(ADP)[31–37], thereby acting as nucleotide-exchange factor (NEF) for both.

The inherent complexity of the triad system has hampered functional understanding. Particularly perplexing is how Ssb engages at the ribosome. It has been cross-linked to ribosomal proteins uL29 and eL39[38]. Yet, these ribosomal proteins are not on the side of the tunnel exit where Zuo1 resides (i.e., near ribosomal proteins eL31 and uL22). Furthermore, uL29/eL39 crosslinking diminishes when J domain function is impaired[38]. So how Zuo1's J-domain is able to engage efficiently with Ssb(ATP) remains unclear, as does the role(s) of Ssz1. We exploited in vivo site-specific cross-linking to obtain snapshots of the positioning of triad components in the cell. Our results point to a pathway of Ssb movement at the ribosome. When in the ATP conformation, Ssb interacts with the NBD of Ssz1. This positioning places it in close proximity to both the J domain of Zuo1 and nascent chains exiting the tunnel. Upon ATP hydrolysis and trapping of the nascent chain, Ssb(ADP) interacts with the ribosome across the tunnel exit, freeing up Ssz1 for binding to another Ssb(ATP).

## Results

**Ssz1 cross-links to Ssb**. To better understand the function of the Ssb:Ssz1:Zuo1 triad system, we carried out in vivo site-specific cross-linking. Using nonsense suppression, the noncanonical, photoactivatable amino acid p-benzoyl-L-phenylalanine (Bpa) was incorporated in place of endogenous amino acids. Starting with Ssz1, we incorporated Bpa at 15 positions distributed across the surface of its NBD. After UV irradiation, cells expressing each Ssz1^Bpa variant were lysed; pelleted ribosomes were subjected to electrophoresis and immunoblot analysis with Ssz1-specific antibody. Species migrating more slowly than Ssz1, either at approximately 180 kDa or 250 kDa, were observed in four cases. Consistent with the known Ssz1–Zuo1 interaction[29], antibodies specific for Zuo1 gave a positive signal for the 180 kDa products observed in cells expressing Ssz1^T217Bpa and Ssz1^Y389Bpa (Fig. 1b).

Because Zuo1 is the J-domain protein partner of Ssb, we tested whether the unidentified 250-kDa products in cells expressing Ssz1^S282Bpa and Ssz1^Y351Bpa were cross-links to Ssb. Both reacted with Ssb-specific antibodies (Fig. 1b). We extended our analysis, carrying out a second round of cross-linking with Ssz1 variants having Bpa at other positions surrounding those that had generated Ssb cross-links. Cross-link products that reacted with Ssb-specific antibodies were observed for seven of these variants (Fig. 1c, Supplementary Fig. 1a, b). Of all the variants, Ssz1^Y351Bpa showed the most robust cross-linking to Ssb. More than half of Ssz1^Y351Bpa cross-linked to Ssb. Cross-linking on this scale suggests that, in the cell, a significant proportion of Ssz1 is interacting with Ssb.

**NBD of Ssz1 interacts with NBD and SBD of Ssb(ATP)**. To further probe the interaction between Ssz1 and Ssb, we carried out, in parallel, unbiased computational docking and analysis of Ssz1^Bpa–Ssb1 cross-linking products using mass spectrometry. Docking yielded a model in which the NBD of Ssz1 interacts with the NBD and the SBDα of Ssb1 when it is in the ATP-bound conformation (Ssb1(ATP)) (Fig. 1d). Mass spectrometry was carried out with two purified Ssz1^Bpa variants, S282Bpa and Y351Bpa, after incubation with Zuo1 and Ssb1 and exposure to UV light (Supplementary Fig. 2, 3). Analysis revealed that Ssz1^S282Bpa cross-linked to Ssb1's NBD peptide [59]NQAALNPR[66], predominantly with Leu63; Ssz1^Y351Bpa cross-linked to SBDα— the extreme Ssb1 C-terminal peptide [609]AMSSR[613]—predominately with the peptide backbone at the very C terminus and the side chains of C-terminal residues A609/M610 (Table 1, Figs. 2, 3). Cross-linking of Ssz1's NBD to both the NBD and SBDα of Ssb1 is consistent with the model presented in Fig. 1d.

We also tested in vivo cross-linking of Ssb1 variants having Bpa incorporated at positions on the NBD and SBDα surfaces predicted to be at the Ssb1(ATP)–Ssz1 interface (Supplementary Fig. 4b). Four variants, two having Bpa in the NBD and two in the SBD, cross-linked to Ssz1 (Fig. 2a). The two NBD variants (Ssb1^Q35Bpa and Ssb1^N59Bpa) showed slowly migrating cross-link

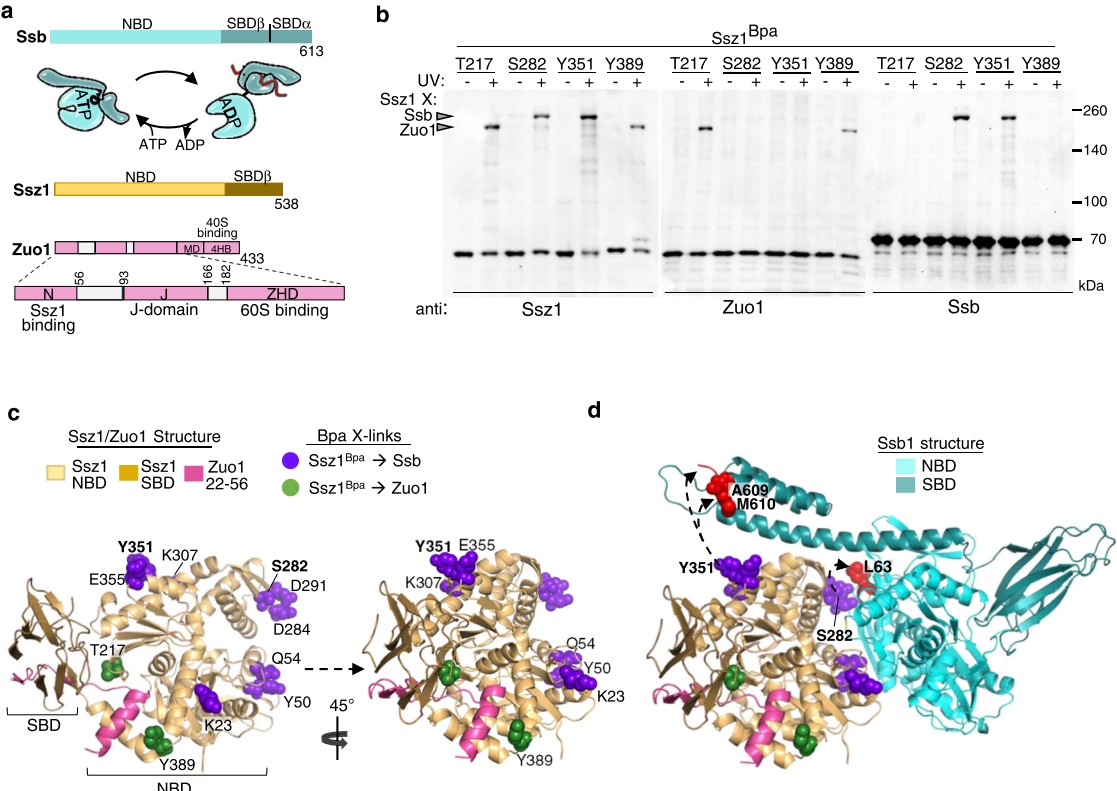

**Fig. 1 The NBD of Ssz1 interacts with the NBD and SBDα of Ssb. a** Schematic of triad components. Line diagrams denoting domains of canonical Hsp70 Ssb, atypical Hsp70 Ssz1 and J-domain protein Zuo1. Cartoon of ATP- and ADP-bound conformations of canonical Hsp70 Ssb1, with substrate bound in SBDβ cleft of Hsp70(ADP) in red. Zuo1 line diagram expanded to emphasize segments functioning at 60 S ribosomal subunit. N domain (N); J-domain (J); Zuotin homology domain (ZHD); Middle domain (MD); 4-helix bundle (4HB). **b** Cross-linking of Ssz1[Bpa] variants. Cells expressing Ssz1 variants with Bpa incorporated at indicated positions were exposed to UV light (+) or left unexposed (−). Cross-linking was analyzed by immunoblotting after SDS-PAGE using antibodies specific for (anti) Ssz1, Zuo1, or Ssb, as indicated. Ssb and Zuo1 cross-link (X) products are indicated with arrowheads; migration of molecular weight markers (kDa) indicated by dashes. Uncropped blots are provided as a Source Data file. Three or more independent strains were analyzed for each Bpa variant, with similar results. **c** Model of *Saccharomyces cerevisiae* Ssz1 bound to residues 22–56 of Zuo1 based on the crystal structure of *Chaetomium thermophilum* proteins (PDB 5MB9). Ssz1 (NBD, beige; SBD, brown), Zuo1 (pink). Positions of Bpa in Ssz1 with identified cross-links shown in sphere representation: to Ssb (purple), to Zuo1 (dark green). See Supplementary Fig. 1 for Ssz1 variants tested. **d** Ssz1–Ssb interaction. The top-ranked model from unbiased docking of Ssz1 and Ssb1 generated using the ZDOCK server starting with *C. thermophilum* Ssz1 (PDB 5MB9) and Ssb1 (PDB 5TKY) crystal structures, then fit to *S. cerevisiae* sequences. Ssz1 is labeled as in (**c**); Ssb NBD, light teal; Ssb SBD dark teal. The peptides of purified Ssb1 that cross-linked to purified Ssz1[Bpa] variants (positions labeled in bold) as determined by mass spectrometry are in red, with predominant cross-linked residues identified in sphere representation and labeled in bold: Ssz1 S282Bpa to Ssb1 L63; Ssz1 Y351Bpa to the C terminus of Ssb1, predominately A609, M610, and the peptide backbone at the extreme C-terminus. See Table 1 and Supplementary Figs. 2, 3 for further information.

**Table 1 Analysis of Ssz1[S282Bpa] and Ssz1[Y351Bpa] crosslinks to Ssb1[a].**

| Ssz1[Bpa] | Ssz1[Bpa] peptide[d] | Ssb1 peptide[d] | Indicated site of Bpa cross-link[e] | Spectral count |
|---|---|---|---|---|
| S282[b] | [272][TLSNATSATIxI][283] | [59][NQAALNPR][66] | L63 (6), P65 (1), A61 (1) | 8 |
| | [271][KTLSNATSATIxI][283] | [59][NQAALNPR][66] | L63 (2), P65 (2) | 4 |
| | [272][TLSNATSATIxIDSLA][287] | [59][NQAALNPR][66] | L63 (3), A61 (1) | 4 |
| | [271][KTLSNATSATIxIDSLA][287] | [59][NQAALNPR][66] | L63 (2) | 2 |
| | [272][TLSNATSATIxI][283] | [251][KTGL][254] | G253 (1) | 1 |
| Y351[c] | [345][LTTNLExTLPESVEILGPQNK][365] | [605][VVTKAMSSR}[613] | C-term (1) | 1 |
| | [345][LTTNLExTLPESVEILGPQNK][365] | [609][AMSSR}[613] | C-term (1), M610 (2), A609 (1) | 4 |

[a]StavroX analysis result depicting dipeptide crosslink products between Ssz1[S282Bpa] or Ssz1[Y351Bpa] and Ssb1.
[b]Cross-linking products were digested with trypsin and AspN.
[c]Cross-linking products were digested with trypsin.
[d]The mass spectrometry data were analyzed using StavroX, with dipeptides listed in the order that they first appear in the ranked StavroX output. Only crosslinked peptides with scores above the false discovery rate of 1% are shown for each experiment. Superscript numbers indicate endpoints of peptide. Brackets "[]" indicate peptide H-N-termini and OH-C-termini, respectively; "}" indicates OH-C-terminus of intact protein. Numbering reflects amino acid numbering of full-length protein; standard one letter designation used except: x = Bpa.
[e]Number in brackets for indicated site of crosslink gives spectral count for that species.

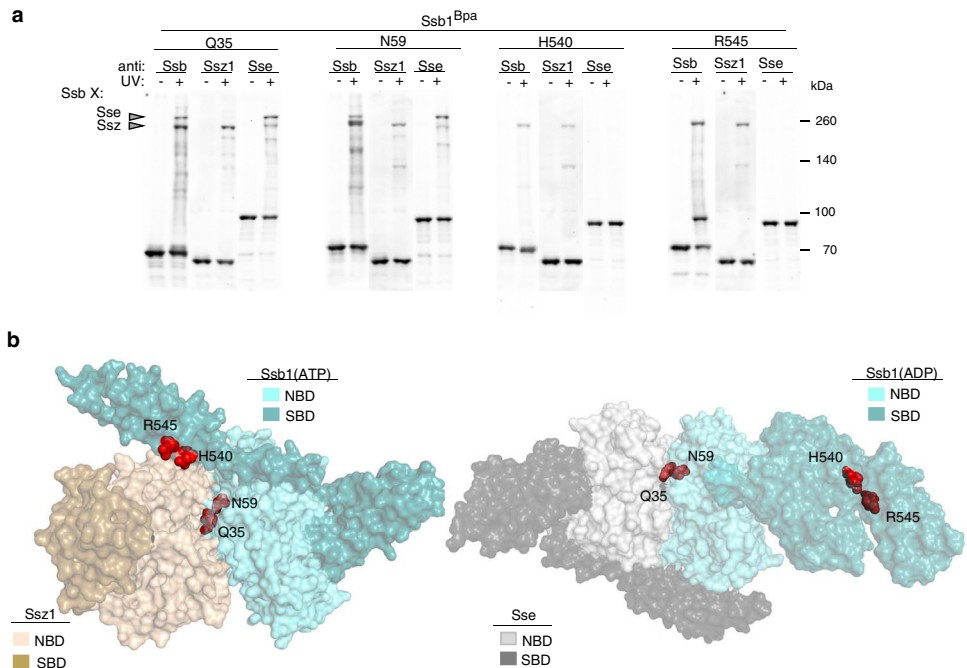

**Fig. 2 Ssb1 cross-links to Ssz1 and Sse. a** Cross-linking of Ssb1[Bpa] variants. Cells expressing Ssb1 with Bpa incorporated at indicated positions were exposed to UV light (+) or left unexposed (−). Cross-linking was analyzed by immunoblotting after SDS-PAGE using antibodies specific for (anti) Ssb, Ssz1, or Sse. Ssb1 cross-links to Ssz1 or Ssb cross-link (X) products are indicated with arrowheads. Migration of molecular weight markers (kDa) is indicated by dashes. Uncropped blots are provided as a Source Data file. Three or more independent strains were analyzed for each Bpa variant, with similar results. **b** Models of Ssb1(ATP)–Ssz1 heterodimer, as in Fig. 1d, and Ssb1–Sse1 heterodimer derived from Sse1-human Hsp70 NBD (PDB 3D2F), and modeled using Ssb1 NBD (PDB 3GL1) and DnaK SBD (PDB 2KHO). Ssb1 positions that cross-linked to either Ssz1 (H540 and R545) or Ssz1 and Sse1 (Q35 and N59) in sphere representation (red).

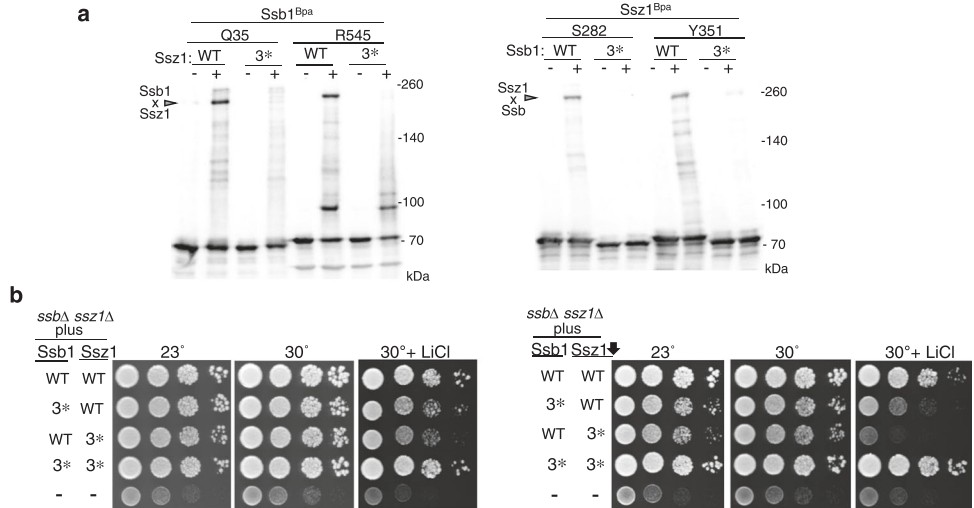

**Fig. 3 Analysis of substitutions at Ssb–Ssz1 interface. a** Cells expressing a Bpa variant of Ssb1 (Q35 or R545) or Ssz1 (S282 or Y351) and either WT or 3* triple-substitution variants of the corresponding partner protein were exposed to UV light (+) or left unexposed (−). Cross-linking was analyzed by immunoblotting after SDS-PAGE with antibodies specific for Ssb. Uncropped blots are provided as a Source Data file. Three independent strains were analyzed for each Bpa variant, with similar results. **b** About 10X serial dilutions of *ssb1-Δ ssb2-Δ ssz1-Δ* cells expressing combinations of WT and 3* Ssb1 (R261D, E370R, and E547R) and 3* Ssz1 variants (R36E, D291R, and R300E) from centromeric plasmids were spotted on rich media containing or lacking 0.1 M LiCl. Ssz1 under endogenous promoter (left) or *CYC1* promoter (right), which provides a lower level of expression, as indicated by downward arrow. Incubation times were three days at 23 °C, two days at 30 °C, three days at 30 °C plus LiCl. Three or more sets of independent strains were analyzed with similar results.

products that did not react with Ssz1-specific antibodies. Since, like other cytosolic Hsp70s, Ssb interacts with the nucleotide-exchange factor Sse, an atypical Hsp70, we tested for Sse cross-reactivity. Finding that the two NBD variants cross-link to Sse as well as Ssb, but the two SBDα variants do not (Fig. 2a), is

consistent with the known mode of Sse1 interaction with canonical Hsp70s (Fig. 2b)—the NBD in the ADP-bound conformation (i.e., Ssb(ADP) interacts with both Sse's NBD(ATP) and its SBDα)[31,32]. This mode is "opposite" to Ssb's interaction with Ssz1—both its ATP-bound NBD and SBDα

interact with the NBD of Ssz1, as indicated by the modeling and cross-linking data reported here.

**Phenotypic consequence of disturbing Ssz1–Ssb heterodimer.** We next undertook a genetic approach to ask if amino acid substitutions at the Ssb–Ssz1 interface affect triad function. We first compared fungal Ssz1 and Ssb1 sequences to identify conserved residues in relatively close proximity at the interface (Supplementary Fig. 4c). For analysis, we chose three charged residues at the interface in each protein that had an oppositely charged residue of its partner protein in relatively close proximity —the rationale being that the resulting charge repulsion would disturb the Ssz1–Ssb1 interaction. Ssz1 residues R36, D291, and R300 were "paired" with Ssb1 residues E370, R261, and E547, respectively. *SSZ1* and *SSB1* mutants, termed *ssz1-3\** and *ssb1-3\**, each containing the three charge-reversal substitutions, encoding Ssz1$^{R36E/D291R/R300E}$ and Ssb1$^{R261D/E370R/E547R}$, were constructed. We then tested the effect of the substitutions on Ssz1/ Ssb1 crosslinking in vivo. Crosslinking of Ssz1$^{S282Bpa}$ to Ssb1–3\* was reduced compared with the wild-type (WT) proteins, as was cross-linking of Ssb1$^{Q35Bpa}$ to Ssz1–3\* (Fig. 3a), indicating disturbance of the Ssz1–Ssb1 heterodimer.

We next asked if these Ssz1/Ssb1 variants could functionally substitute for WT proteins in vivo. In initial tests, the two variants supported growth nearly as well as WT at both 23 and 30 °C, with modest growth defects observed in the presence of LiCl (Fig. 3b). We then tested the effect of the 3\* mutations when the level of Ssz1 was reduced, as we previously showed that low level of expression of Ssz1 driven by the *CYC1* promoter was sufficient for robust cell growth[22]. Growth phenotypes were observed for both 3\* variants, not only in the presence of LiCl, but also at 23 °C. As a test of whether the phenotypes observed were directly related to the charge reversals, we compared the growth of cells expressing both variants to that of cells expressing one variant and one WT partner. Cells expressing both variants grew better (Fig. 3b), indicating that neutralization of the charge repulsion occurring upon expression of a single variant promoted better growth, and supporting the idea that the Ssz1–Ssb(ATP) interaction plays a functional role in vivo. As considered more thoroughly in the discussion, this interaction of Ssb(ATP) with Ssz1, coupled with known attributes of Zuo1/Ssz1 architecture, also places constraints on its location at the ribosome, near the interaction site of Zuo1 (and hence Ssz1).

**Ssb(ADP) positioning at the ribosome.** It has been reported, using nonspecific cross-linkers, that Ssb cross-links to ribosomal proteins uL29 and eL39, and that the amount of cross-linking is diminished when the Zuo1 J-domain is inactivated[38]. Because cross-linkers incorporated at single positions enable more specific positioning of interacting proteins, we decided to use Bpa to better define Ssb-ribosomal protein interactions. Having noted a strong, rapidly migrating Ssb cross-link product (at ~90 kDa) in the case of Ssb1$^{R545Bpa}$ (Fig. 2a), we tested for reactivity with antibodies specific for ribosomal proteins near the exit site— specifically eL19, eL31, uL22, uL24, eL39, uL29, and uL23. A positive signal was generated with uL29-specific antibodies (Fig. 4a, Supplementary Fig. 5a). Next, we employed a Zuo1 variant (Zuo1$^{H128Q}$) having a substitution in the invariant HPD motif of the J-domain, which disables its ability to stimulate Ssb's ATPase activity[27]. The Ssb1–uL29 cross-link product was much less abundant in *zuo1$^{H128Q}$* cells than in *ZUO1* cells (Fig. 4a), while Ssz1–Ssb cross-links were similar in the two strain backgrounds (Fig. 4a, Supplementary Fig. 5b).

When considering these results, we noted that when Ssb is in the ADP-bound conformation, R545 is located on the same SBD

face as SBDβ residues $^{428}$KRR$^{430}$, previously shown to be important for stable Ssb association with the ribosome[39]. Therefore, we asked if Bpa, when positioned on this SBDβ face, could be cross-linked to ribosomal proteins. Bpa was incorporated at 5 positions (Supplementary Fig. 5c). The variants showed a "background smear" of cross-linking, likely due to the proximity of Bpa to the peptide-binding cleft. However, one (Ssb1$^{K497Bpa}$) formed a cross-link product that migrated at ~90 kDa. The abundance of this cross-link product, which reacted with uL24-specific antibodies, was diminished in *zuo1$^{H128Q}$* cells (Fig. 4b, Supplementary Fig. 5a). As uL24 had not, to our knowledge, been previously identified as a site of Ssb1 cross-linking, we carried out a "reverse" cross-linking experiment, incorporating Bpa into surface-exposed positions of uL24, and as a control, uL29. Two uL29 variants (E27Bpa and R38Bpa) and two uL24 variants (K84Bpa and E88Bpa) cross-linked to Ssb (Fig. 4c; Supplementary Fig. 6). In all four cases, cross-linking to Ssb was less in cells expressing Zuo1$^{H128Q}$ than in cells expressing WT Zuo1 (Fig. 4c).

Together, these results are consistent with Ssb cross-linking to uL24 and uL29 when in the ADP-bound conformation. The distance between the residues of uL24 and uL29 in the ribosome that cross-link to Ssb is compatible with the distance between the residues in SBDβ and SBDα, K497Bpa and R545Bpa, respectively, in Ssb1(ADP). Positioning based on these constraints places the SBD of Ssb(ADP) on the ribosome with the tip of the SBDα lid and loops 3/4 and 5/6 of SBDβ pointing toward the tunnel exit (Figs. 4c, 5), and the NBD away from it. Such placement suggests that the KRR interacts either with uL24 or the rRNA between it and uL29.

## Discussion
Data presented here, together with previously published information, point to a pathway of Hsp70 action at the ribosome (Fig. 5): in the ATP-bound conformation, canonical Hsp70 Ssb interacts with atypical Hsp70 Ssz1, positioning it in close proximity to the J-domain of cochaperone partner Zuo1; upon ATP hydrolysis, which stabilizes its binding to nascent chain, Ssb interacts directly with the ribosome. The unanticipated finding that ATP-bound Ssb interacts with the NBD of Ssz1 is supported by cross-linking analyses, as well as unbiased computational docking and the persistence of the interaction when Zuo1's J-domain is inactivated. Such Hsp70–Hsp70 interactions are likely ancient. Indeed, *Escherichia coli* Hsp70 DnaK molecules, in the ATP-bound conformation, form homodimers, with each NBD interacting with both the NBD and SBDα of its binding partner[40]. Gene duplications likely allowed for specialization of such interactions over time, resulting in the emergence of other activities such as that of Ssz1 uncovered here, as well as that of the NEFs of the eukaryotic cytosol and ER lumen[26,41].

Uncovering Ssz1–Ssb(ATP) heterodimerization, together with being able to position Ssb(ADP) relative to uL24/29 helps clarify a somewhat confusing picture of Ssb at the ribosome. We conclude that Ssb occupies two distinct positions, with a shift of the entire Ssb molecule across the tunnel exit, triggered by the hydrolysis of ATP. The architecture of Zuo1, particularly its interaction with Ssz1 via its N-terminal segment and with the 60 S subunit (near eL31 and uL22) via its ZHD, is key to placement of Ssb(ATP) (presented in more detail in Fig. 5). The most stringent constraint comes from the fact that the J-domain must, at least transiently, interact at Ssb's NBD/SBDβ interface, yet only 23 residues separate the ZHD from the J-domain (and only 37 residues between the J-domain and the Ssz1-interacting segment). Ssb(ADP) positioning is more straightforward because of the availability of high-resolution ribosome structures. Our cross-

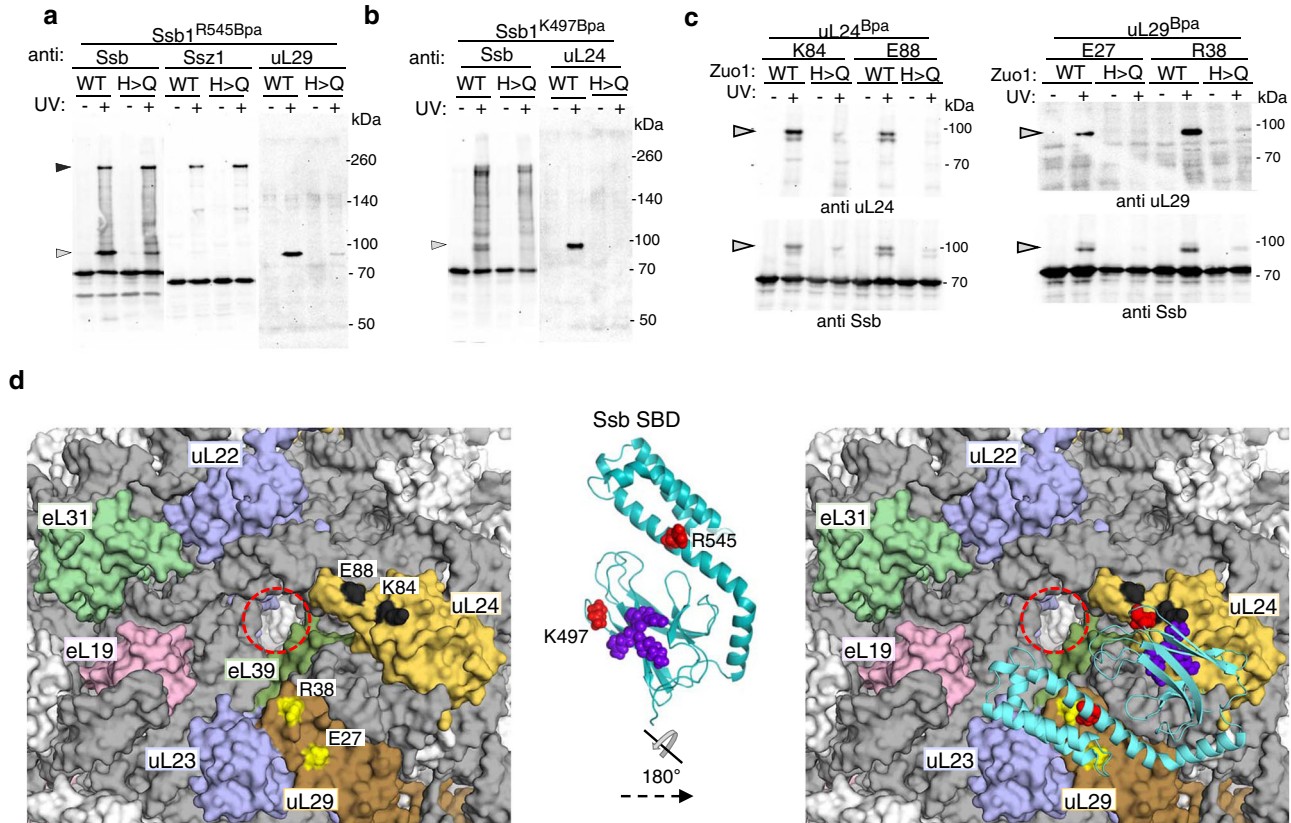

**Fig. 4 Ssb cross-linking to Ssz1 and ribosomal proteins in *zuo1*[H128Q] background. a, b** *ZUO1* (WT) and *zuo1*[H128Q] (H > Q) cells expressing Ssb1[R545Bpa] (**a**) or Ssb1[K497Bpa] (**b**) were exposed to UV light (+) or, as a control, left unexposed (−). Cross-linking was analyzed by immunoblotting after SDS-PAGE with antibodies (anti) specific for indicated proteins. Arrowheads indicate cross-link products between Ssb1 and ribosomal protein uL29 or uL24. Uncropped blots are provided as a Source Data file. Three independent strains were analyzed for each Bpa variant, with similar results. **c** *ZUO1* (WT) and *zuo1*[H128Q] (H > Q) cells expressing Bpa variants of uL24 (K84 or E88) or uL29 (E27 or R38) were exposed to UV light (+) or, as a control, left unexposed (−). Cross-linking was analyzed by immunoblotting after SDS-PAGE with antibodies (anti) specific for indicated proteins. Arrowheads indicate cross-link products between uL24 or uL29 and Ssb. Uncropped blots are provided as a Source Data file. Three independent strains were analyzed for each Bpa variant, with similar results. **d** *left*, structure of *S. cerevisiae* 60 S ribosome subunit (PDB 3J78). Ribosomal proteins near the tunnel exit, which were tested in this study, are colored and labeled; ribosomal proteins not tested in dark gray; ribosomal RNA in light gray. Red dashed circle surrounds the tunnel exit. Positions of uL29 and uL24 that cross-linked to Ssb shown in yellow (R38, E27) and black (K84, E88), respectively. SBD of Ssb1(ADP), modeled using the DnaK SBD structure (PDB 2KHO), with residues (428–430) of [428]KRR[430] tripeptide (purple) and positions cross-linking to ribosomal proteins when Bpa incorporated (R545 and K597, red) shown in sphere representation. *right*, SBD flipped 180° and positioned on the ribosome structure in PyMOL manually according to cross-linking data shown in (**a–c**).

linking results are consistent with interaction of SBDα and SBDβ with ribosomal proteins uL29 and uL24, respectively—with the SBDβ tripeptide (KRR) important for Ssb ribosome association, interacting either with uL24 or rRNA between the two ribosomal proteins[39]. Although we did not observe cross-linking of Ssb1[Bpa] variants to eL39, as reported for nonspecific cross-linking[38], it is in very close proximity to SBDβ of Ssb in the model presented here.

The exact functions of the Ssz1–Ssb interaction itself, and the transition to direct ribosome binding remains unresolved. However, one advantage of Ssb interaction with Ssz1 is apparent—positioning Ssb(ATP) both near the exit of the ribosome tunnel for interaction with nascent chain substrates and in close proximity to the J-domain of Zuo1, whose transient interaction at the Ssb's NBD–SBDβ interface is required. But, the machinery may be even more complex. Recent analysis using in vitro translation systems indicated that Ssz1's peptide-binding cleft interacts with nascent chains, displacing the very N-terminal residues of Zuo1[30]. Extremely short nascent chains bind in Ssz1's peptide-binding cleft, with Ssb interacting with slightly longer ones—suggesting that Ssz1 interacts first with emerging nascent chains, and Ssb

second. However, Ssz1–Ssb heterodimerization presents a challenge in placing SBDβs of both Hsp70s in close proximity to nascent chains exiting the tunnel. We speculate that the Ssz1–Ssb heterodimer may not be static. In addition to the 23 residues between the end of Zuo1's J-domain and ZHD, there are 37 residues, whose structure is unresolved, between the end of the segment that stably interacts with Ssz1 and the beginning of the J-domain. This is more than sufficient to bridge the distance for J-domain interaction at Ssb's NBD–SBDβ interface[42] and may also allow dynamic regulatory rearrangements.

In the case of the transition across the exit, the simplest explanation is that this change in position "frees up" Ssz1 for binding another Ssb(ATP) for engagement with the emerging nascent chain, thus potentially allowing two Ssb molecules to be at a ribosome simultaneously (Fig. 5). More speculatively, Ssb movement may facilitate nascent chain advancement out of the tunnel. Such action would be analogous to the function of Hsp70 of the mitochondrial matrix "import motor" in driving translocation of proteins through the channel in the inner membrane[43,44]. Though such a role for Ssb would be at a much more limited scale, as the translation process and the nascent

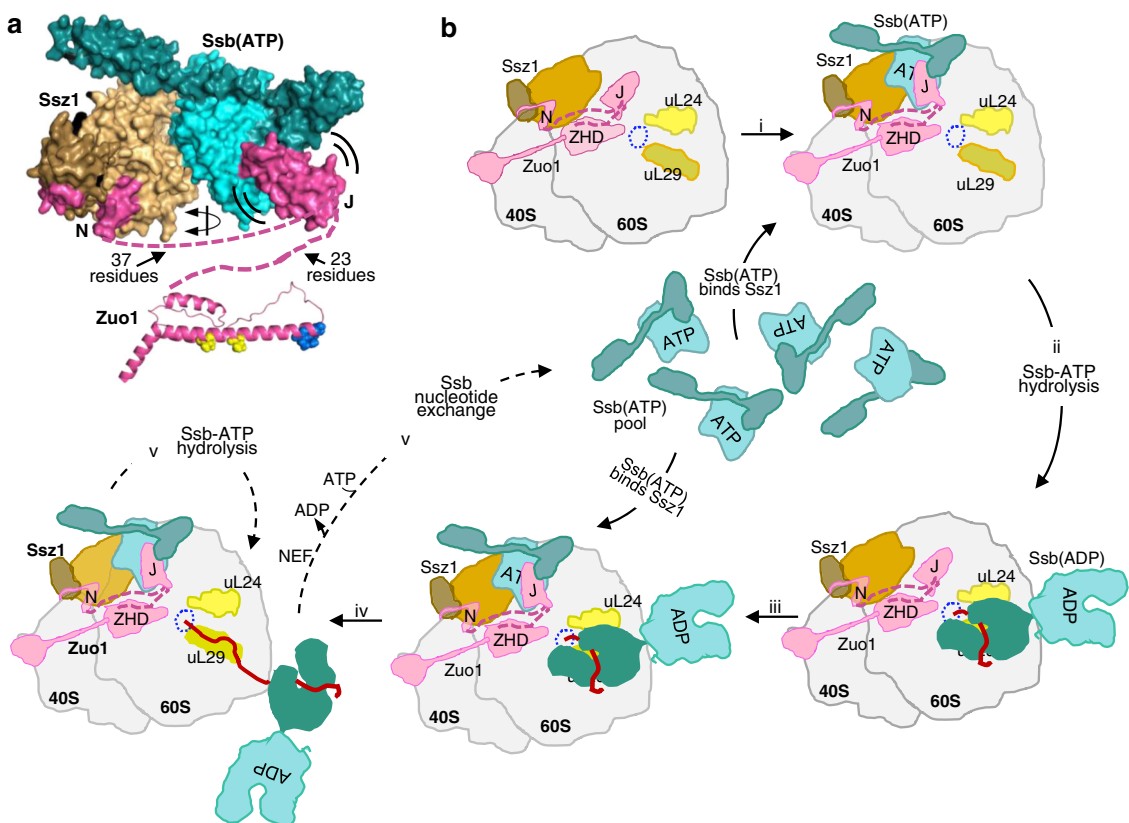

**Fig. 5 Model of Ssb interaction at the ribosome. a** Constraints placed by Zuo1 (pink) and Ssz1 (brown) on Ssb1(ATP) (teal) positioning at 60 S subunit. Ssz1, Ssb(ATP), and regions of Zuo1 that directly interact with the heterodimer shown in surface representation. Double-arced lines around J-domain indicate that interaction with Ssb's NBD–SBDβ interface is transient. Structure of Zuotin homology domain (ZHD) (PDB 5DJE) shown in cartoon representation; residues involved in 60 S ribosome interaction in sphere representation (eL31 interaction, yellow; R247/251 thought to interact with ribosomal RNA helix 24, blue). Dashed pink lines indicate the segment of Zuo1 whose structure is unknown: 37 residues between the N segment and the J-domain; 23 residues between the J-domain and beginning of ZHD. The latter constrains the distance between Zuo1 and Ssb(ATP) because of requirement for J-domain interaction. Double-arrowed line indicates that the exact orientation of Ssz1 NBD, and hence Ssb1(ATP), is not known because structures of the region of Zuo1 indicated by dashed lines are not known. J-domain (J) residues 93–166 (modeled using the J-domain of Ydj1, PDB 5VSO) and positioned at the NBD/SBDβ interface of Ssb1(ATP) based on the crystal structure of DnaJ J-domain complexed with DnaK (PDB 5NRO). **b** Schematic of the pathway of Ssb interaction at the ribosome. Top left: Ribosome, gray; tunnel exit, dotted blue circle; Ssb1, teal, as in (**a**); Zuo1, pink, as in (**a**) and in addition, MD and 4HB interacting with the 40 S subunit; ribosomal proteins uL24 and uL29 in light and dark yellow, respectively; Ssz1: NBD, light brown; SBD, dark brown. (i) Ssb(ATP) binds Ssz1. (ii) Upon hydrolysis of bound ATP, stimulated by J-domain and nascent chain, Ssb changes position, with SBD locating at uL29 and uL24, freeing up space for (iii) binding of another Ssb(ATP) to Ssz1. (iv) Nascent chain extension proceeds and Ssb(ADP) moves away from the ribosome; at some point (v) a NEF facilitates the exchange of ATP for ADP on Ssb, resulting in release of nascent chain and return of Ssb to the Ssb(ATP) pool; Ssb1(ATP) bound to Ssz1 occupies the uL29/uL24 position after ATP hydrolysis.

chain itself provide force for nascent chain movement[45,46]. Nevertheless, it is intriguing that selective ribosome-profiling experiments revealed that Ssb binding coincided with faster translation rates[47]. Extensive analysis of translation of specific nascent chains will be required to test this idea.

The results presented here advance our understanding of the function of the triad chaperone system. However, one can easily envision that these interactions among the triad components themselves and the nascent chain—Zuo1 with both Ssz1 and Ssb, and Ssz1 with nascent chains—could be fine-tuned to serve as a regulatory mechanism for optimizing Ssb interaction with the nascent chain. In this regard, it should be noted that in the absence of Zuo1–Ssz1, Ssb still interacts with nascent chains in vivo, albeit less efficiently, and with less specificity[47,48]. In addition, the intriguing relationship between the interaction of the C-terminal domain of Zuo1 with the 40 S subunit and translational fidelity[20,49] places the triad system at the nexus of protein translation and protein folding. More extensive work is required to fully understand these mechanisms, including

comparative studies across organisms. A more sophisticated understanding of this complex molecular chaperone machinery may also serve as a roadmap to better understand other, less well-studied Hsp70-based machineries, and chaperone machineries in general.

## Methods

**Yeast strains and plasmids.** Lists of *S. cerevisiae* plasmids and strains used in this study are shown in Supplementary Tables 1 and 2, respectively. Codon-substitution mutant genes were constructed in plasmids by Quick Change PCR mutagenesis (Stratagene) using oligonucleotides listed in Supplementary Table 3. Mutations were verified by DNA sequencing. All yeast strains used were of the DS10 genetic background[24]. Plasmids pRS316-uL24b and pRS316-uL29a were constructed by amplification of the respective genes from genomic DNA, followed by cloning into Bam HI-digested pRS316 DNA using the HiFi DNA assembly kit (New England Biolabs, Inc.). KanMX disruptions in *SSB1*, *ZUO1*, *RPL26B* (encoding uL24b), and *RPL35A* (encoding uL29a) were generated by polymerase chain reaction (PCR) amplification of the KanMX-disrupted gene locus[50], followed by transformation of the desired DS10 parent strain to obtain KanR colonies. Replacement of the gene with the KanMX-disrupted locus was confirmed by PCR analysis of genomic DNA in each case.

**In vivo cross-linking**. All yeast strains to be used for in vivo crosslinking were transformed with the ptRNA-Bpa plasmid, which directs incorporation of Bpa at TAG codons, and a plasmid having a TAG mutation at the desired position for Bpa incorporation in the target protein. Such yeast strains were grown on the appropriate selective minimal medium with 2 mM Bpa, starting at an $OD_{600nm}$ between 0.03 and 0.10. At $OD_{600nm} = 1.7–2.1$, cycloheximide was added to 0.1 mg/ml and cells harvested by centrifugation. In total, 40–80 $OD_{600nm}$ units of resuspended cells were divided in half—one half subjected to 365-nm UV illumination for 1 h at 4 °C (Stratalinker 1800 UV cross-linker), while the other half was kept on ice as a control. Cells were lysed by agitation with glass beads for 5 min at 4 °C in lysis buffer (300 mM sorbitol, 20 mM HEPES-KOH, pH7.5, 1 mM EGTA, 5 mM MgCl₂, 10 mM KCl, 10% glycerol v/v, 1 mM dithiothreitol, and RNasin RNase Inhibitor (Promega) at a dilution of 1:1000). Lysates were clarified by centrifugation at 16,100 xg for 10 min in a microcentrifuge (Eppendorf). In total, 5 $OD_{260}$ units of cleared cell lysate were loaded onto a 0.8-ml sucrose cushion composed of lysis buffer with 0.5 M sucrose instead of sorbitol. Ribosomes were pelleted by centrifugation for 2 hr at 150,700 xg in a MLA130 rotor (Beckman Coulter) at 4 °C. The ribosome-containing pellet was suspended in SDS sample buffer (0.124 M Tris-HCl buffer, pH 6.8, 4% w/v SDS, 10% v/v glycerol, 0.02% bromophenol blue, and 4.5% β-mercaptoethanol) and used for immunoblotting as described below.

All immunoblot analyses were carried out using the Enhanced Chemiluminescence system (GE Healthcare, UK, Buckinghamshire, UK) according to the manufacturer's instruction. Uncropped blots are shown in the Source data file. Rabbit polyclonal antibodies were used throughout, using ECL anti-rabbit IgG (Amersham) as secondary for detection at 1:40,000 dilution. Antibodies specific for Ssb1[51] (1:3000), Ssz1[18] (1:2000), Zuo[20] (1:1000), and Sse1 (1:1000) were used at dilutions in parentheses. Sse1 antibody was prepared by immunizing rabbits with purified Sse1 fused at the C terminus with the purification tag, glutathione-S-transferase, which was used for purification by binding to glutathione and verified by testing with extracts from a strain lacking the *SSE1* gene (Supplementary Fig. 4). Antibodies specific for ribosomal proteins eL19 (1–7000), uL22 (1:10,000), uL23 (1:10,000), uL24 (1:10,000), uL29 (1:20,000), and eL39 (1:5000) were kindly provided by Sabine Rospert[17,52] and used at dilutions in parentheses. Polyclonal anti-HA antibodies (Proteintech Group, Inc.) were used at a 1:2000 dilution to detect eL31, which has a HA tag on its C-terminus[20].

**Modeling and docking**. Model structures were generated using Modeller[53], then imported into PyMOL (The PyMOL Molecular Graphics System, Version 2.4 Schrödinger, LLC) for viewing, alignment, and generation of images for figures. The ZDOCK server[54] was employed for docking. The crystal structures of Ssz1 (PDB 5MB9) and ATP-bound Ssb1 (PDB 5TKY) of *C. thermophilium* were used as input without contacting or blocking residue selection. To build a *S. cerevisiae* Ssz1 and Ssb1 docking model, the modeled Ssz1 and Ssb1 *S. cerevisiae* structures were aligned individually to those in the top-ranked *C. thermophilium* modeled heterodimer. To aid in selection of conserved residues to change for genetic experiments, fungal Ssb and Ssz1 sequences[55] were analyzed using this model and the CONSURF server[56,57].

**Protein expression, purification, and in vitro cross-linking**. Open-reading frames of *SSB1*, *SSZ1*, and *ZUO1* were modified by PCR to introduce Bam HI and Xho I sites at the 5′ and 3′ ends, respectively, and cloned into plasmid pSmt3 to generate N-terminal His₆-Smt3 (SUMO) fusions[58]. In the case of Ssz1, the native amber stop codon (TAG) stop codon was replaced with TGA and the codon for S282 and Y351 was replaced by TAG to enable Bpa incorporation. For Ssb1 and Zuo1, the resulting expression plasmids were used to transform *E. coli* BL21(DE3) pLysS. The Ssz1 plasmids were transformed into *E. coli* BL21 along with suppressor vector pSupT/BpF[59,60]. Saturated cultures of the expression strains were diluted to $OD_{600nm} = 0.1$ in LB medium (supplemented with 200 mg/l arabinose in the case of Ssz1-expression strains) and incubated at 37 °CC with shaking. At $OD_{600nm}$ of approximately 0.6, cultures were chilled to 18 °CC and IPTG added to a final concentration of 0.5 mM. In the case of Ssz1-expression cultures, Bpa was also added at a final concentration of 1 mM. Incubation was continued overnight at 18 °C.

Cells were lysed by suspension in lysis buffer (25 mM HEPES, pH 7.5, 150 mM NaCl, 30 mM KCl, 10 mM MgOAc, 25 mM imidazole, and 1 mM DTT) and, for Ssb1 only, 1 mM ADP, followed by using either a French Pressure Cell (American Instrument Company) or Cell Disrupter (Constant Systems LTD) to lyse the cells. Lysates were clarified by centrifugation at 20,000 xg for 30 min, and purified using HisPur Ni-NTA Resin (Thermo Scientific) as recommended by the manufacturer. The fusion tag was cleaved by treatment with His₆-tagged Ulp protease[58]. The mixture was desalted on a NAP-10 column (GE Healthcare) and the desired product purified by a second passage over HisPur Ni-NTA Resin (Ssz1, Zuo1). A final step of ATP agarose (Sigma-Aldrich) affinity purification was used for Ssb1[61].

For in vitro cross-linking reaction mixtures, each protein was at a concentration of 5 mM in reaction buffer (10 mM Tris 7.5, 50 mM KCl, 5 mM MgCl₂, 1 mM ATP, and 1 mM dithiothreitol). Reaction mixtures in half-area 96-well plates (Greiner Bio-One) were subjected to 365-nm UV illumination for 30 min at 4 °CC (Stratalinker 1800 UV cross-linker), while control reactions were kept on ice. Samples of the reactions were combined with sample buffer as described above and

resolved on SDS-PAGE gels (7.5% acrylamide, unless otherwise indicated) with Coomassie staining used to visualize bands.

**Mass spectrometry sample preparation**. Polyacrylamide gels were stained with Coomassie R250 to visualize protein bands. Gel pieces were destained completely in MeOH/H₂O/NH₄HCO₃ (50%/50%/100 mM), dehydrated for 2 min in ACN/H₂O/NH₄HCO₃ (50%/50%/25 mM), and then once more for 30 sec in 100% ACN. After destaining, all samples were dried in a Speed-Vac for 1 min, reduced in 25 mM DTT (dithiothreitol in 25 mM NH₄HCO₃) for 15 min at 56 °C, alkylated with 55 mM IAA (iodoacetamide in 25 mM NH₄HCO₃) in darkness at room temperature for 15 min, washed once in H₂O, dehydrated for 2 min in ACN/H₂O/NH₄HCO₃ (50%:50%:25 mM), and then once more for 30 sec in 100% ACN. Samples were dried and rehydrated with 20 µl of trypsin solution with 0.01% ProteaseMAX™ surfactant (10 ng/µl *Trypsin* from Promega Corp. in 25 mM NH₄HCO₃/0.01% w/v of ProteaseMAX™ from Promega Corp.). Samples were allowed to stand for 2 min at room temperature prior to addition of an additional 30 µl of overlay solution (25 mM NH₄HCO₃/0.01% w/v of ProteaseMAX™), which was added to keep gel pieces immersed throughout the digestion. The digestion was conducted for 3 h at 42 °C. For all samples, proteolysis was terminated by acidification with 2.5% TFA (trifluoroacetic acid) to 0.3% final (10 µl added). Degraded ProteaseMAX™ was removed via centrifugation (max speed, 10 min) and the peptides solid phase extracted (Bond Elut *OMIX C18* pipette tips from Agilent). Peptides were eluted off the C18 column with 15 µl of acetonitrile/H₂O/TFA (70%:30%:0.1%) dried to minimum volume and then brought up to 20 µl of total volume with 0.1% formic acid; 3 µl was loaded on the instrument.

For further digestion of Ssz1$^{S282Bpa}$ in vitro cross-link products, soluble peptides generated from tryptic digestion were transferred to a new Protein LoBind tube (~50 µl volume) and secondary digestion carried out using 5 µl of AspN proteinase (20 ng/µl endoproteinase AspN from Roche Diagnostics in 25 mM NH₄HCO₃) for 2 h at 37 °C. Concurrently, the remaining gel pieces post tryptic digestion were dehydrated with 10 µl of ACN and rehydrated with 20 µl of secondary digestion mix composed of 5 µl of AspN proteinase (20 ng/µl endoproteinase AspN from Roche Diagnostics in 25 mM NH₄HCO₃), 13 µl of 25 mM NH₄HCO₃, and 2 µl of 0.1% ProteaseMAX™ surfactant. This digestion was allowed to proceed for 2 h at 37 °C, after which the solution was removed and combined with the primary "in liquid" AspN digestion. Proteolysis was terminated and subsequent steps performed as described above.

**NanoLC–MS/MS, data analysis, and cross-linking assignment**. Peptides were analyzed by nanoLC–MS/MS using the Agilent 1100 nanoflow system (Agilent) connected to a hybrid linear ion trap–orbitrap mass spectrometer (LTQ-Orbitrap Elite™, Thermo Fisher Scientific) equipped with an EASY-Spray™ electrospray source, essentially as described in Nguyen et al.[62]. MS scans were acquired in the Orbitrap with a resolution of 120,000 followed by MS2 for the 20 most intense peptides detected in the MS1 scan from 350 to 1800 m/z; redundancy was limited by dynamic exclusion with repeat count of 2- and 15-s duration. CID-based MS/MS fragmentation in the ion trap portion of the instrument used activation time of 10msec, normalized collision energy of 35, and isolation width of 2AMU. Monoisotopic precursor selection and charge-state screening were enabled and +2 and lower charge states were rejected. To obtain identifications of un-cross-linked peptides raw MS/MS data were converted to mgf file format using MSConvert (ProteoWizard: Open Source Software for Rapid Proteomics Tools Development). The resulting mgf files were used to search against *E. coli* amino acid sequence database containing Ssz1 constructs with decoy reverse entries and a list of common contaminants (8493 sequence entries) using the UW Biotechnology Center *Mascot* search engine 2.2.07 (Matrix Science) with variable methionine oxidation with asparagine and glutamine deamidation plus fixed cysteine carbamidomethylation. To determine the identity of cross-linked products, a targeted database search of raw mass spec data was conducted using only the sequence of proteins used in the crosslinking experiment plus common lab protein contaminants. Cross-linked candidates were generated using StavroX software (StavroX Freeware version 3.6.6. from University of Halle–Wittenberg) with BPA chosen as a cross-linker, trypsin and AspN as protease digestion sites with static cysteine carbamidomethylation, plus variable methionine oxidation as possible modifications. Peptide mass tolerance was set at 10 ppm and fragment mass at 0.6 Da. All crosslinking candidates were below 1% of the false-discovery rate. Designation of cross-linking sites was based on the number of occurrences of the cross-linking sites of the candidates within the top-ten highest scores. MS1 and MS/MS spectrum of cross-linking candidates was manually examined. Cross-linking results were cross-checked with UV-unexposed or individually UV-exposed proteins as controls to ensure robustness.

**Reporting summary**. Further information on research design is available in the Nature Research Reporting Summary linked to this article.

## Data availability

The data that support the findings of this study and biological materials are available from the corresponding authors on reasonable request. The mass spectrometry proteomics data have been deposited to the ProteomeXchange Consortium via the

PRIDE[63] partner repository with the dataset identifier PXD024065. Previously published structures used for modeling are available from the PDB under accession codes 5MB9 [https://doi.org/10.2210/pdb5MB9/pdb], 5TKY [https://doi.org/10.2210/pdb5TKY/pdb], 3D2F [https://doi.org/10.2210/pdb3D2F/pdb], 3GL1 [https://doi.org/10.2210/pdb3GL1/pdb], 2KHO [https://doi.org/10.2210/pdb2KHO/pdb], 3J78 [https://doi.org/10.2210/pdb3J78/pdb], 5DJE [https://doi.org/10.2210/pdb5DJE/pdb], 5VSO [https://doi.org/10.2210/pdb5VSO/pdb], and 5NRO [https://doi.org/10.2210/pdb5NRO/pdb]. Source data are provided with this paper.

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

## Acknowledgements

We thank Szymon Ciesielski for helpful comments on the paper, and Sabine Rospert for antibodies to ribosomal proteins. This work was funded by National Institutes of Health grant R01 GM31107 and R35 GM127009 (EAC).

## Author contributions

K.L, T.Z., W.D. and S.E.B. performed in vivo cross-linking experiments and construction of mutants required. K.L. and T.Z. performed genetic analyses and in vitro cross-linking. G.S. performed mass spectrometry. E.A.C., K.L. and T.Z. wrote the paper. All authors commented on the paper.

## Competing interests

The authors have no competing interests.
