## [Peer Review File · Nature Communications]

Pathway of Hsp70 interactions at the ribosomeReviewers' Comments:

Reviewer #1:

Remarks to the Author:

Nascent polypeptides emerging from the translating ribosome encounter the cellular protein folding machinery. In eukaryotes this involves the Hsp70/Ssb and atypical Hsp70-Ssz1 system together with co-chaperone/J-domain containing Zuo1. While individual protein:protein interactions of this triad has been characterized by structural studies, their interaction dynamics in vivo remains mysterious. By employing site specific cross-linking chemistry in vivo, the Craig group now provides insight into the intricate/ordered interplay between the Ssb, Ssz1 and Zuo1 and the nascent chain during protein bio-genesis. The work is of general interest to the protein folding chaperone fields and attempts to bridge static pictures obtained through crystallography/cryo-EM with the in vivo situation. Overall, the work has been carefully planned/carried out and interpreted well. The experiments presented are systematic and support the overall conclusions. I do not have any complaints!
I support publication of this study in Nat. Comm.

Reviewer #2:

Remarks to the Author:

In this manuscript, the authors utilized in vivo site-specific crosslinking and mapped out subunit interactions in yeast Hsp70 chaperone triad. Two distinct modes of yeast Hsp70 Ssb1 interactions regulated by ATP hydrolysis elucidated a novel pathway of Hsp70 action at the ribosome. Mutagenesis at the Ssb1/Ssz1 interface and growth assay validated crosslinking data in ATP bound state. Reverse crosslinking and data from a Zuo1 ATPase activation defect mutant corroborated the ADP conformation model. Overall, the biochemical and functional characterization of yeast Hsp70 chaperone triad seems thorough with multiple approaches and the scope suitable for Nature Communications. However, some experimental details need to be clarified and discussions included to ensure scientific rigor:

1. The mass spectrometric data on Ssz1 Y351Bpa crosslinking (Supplementary Fig. 3) is weak. The absence of fragments from the proposed Ssb1 peptide with an Arginine is surprising (S.F. 3B). The assignment of Ssb1 peptide was essentially only based on mass, which is not likely conclusive for trypsin and Asp-N double digestion. There was not a compelling separation between the candidates and decoys in StavroX decoy analysis. The proposed Ssb1 crosslinked peptide has a miscleavage site. Was the completely digested version of this crosslink pair detected? How about the version of peptide without Methionine oxidation? If there is no additional MS data to support this assignment, the authors should include relevant discussions and probably justify why Ssb1 R540 and R545 were chosen for later study presented in Fig. 2.
2. The comparison of Ssb1/Ssz1 and Ssb1/Sse1 interactions was confusing both in text and Fig. 2B. The position of Ssb1 NBD should be consistent in both images to illustrate the difference in ATP and ADP state, as well as interaction surfaces. Similarly, it will be visually helpful to align Ssb1 and Ssz1 as how they form the complex in Sup Fig. 4C for readers to appreciate the charge complementarity.
3. For MS data acquisition in Method section, resolution, mass isolation window and mass analyzer for MS1 and MS2 data should be specified and matched to the search parameters in Stavrox.

Response to reviewers' comments:

REVIEWER 1

Nascent polypeptides emerging from the translating ribosome encounter the cellular protein folding machinery. In eukaryotes this involves the Hsp70/Ssb and atypical Hsp70-Ssz1 system together with co-chaperone/J-domain containing Zuo1. While individual protein:protein interactions of this triad has been characterized by structural studies, their interaction dynamics in vivo remains mysterious.

By employing site specific cross-linking chemistry in vivo, the Craig group now provides insight into the intricate/ordered interplay between the Ssb, Ssz1 and Zuo1 and the nascent chain during protein bio-genesis. The work is of general interest to the protein folding chaperone fields and attempts to bridge static pictures obtained through crystallography/cryo-EM with the in vivo situation. Overall, the work has been carefully planned/carried out and interpreted well.

The experiments presented are systematic and support the overall conclusions. I do not have any complaints!

I support publication of this study in Nat. Comm.

Response: We thank the reviewer for the supportive comments.

REVIEWER 2

In this manuscript, the authors utilized in vivo site-specific crosslinking and mapped out subunit interactions in yeast Hsp70 chaperone triad. Two distinct modes of yeast Hsp70 Ssb1 interactions regulated by ATP hydrolysis elucidated a novel pathway of Hsp70 action at the ribosome. Mutagenesis at the Ssb1/Ssz1 interface and growth assay validated crosslinking data in ATP bound state. Reverse crosslinking and data from a Zuo1 ATPase activation defect mutant corroborated the ADP conformation model. Overall, the biochemical and functional characterization of yeast Hsp70 chaperone triad seems thorough with multiple approaches and the scope suitable for Nature Communications. However, some experimental details need to be clarified and discussions included to ensure scientific rigor:

1. The mass spectrometric data on Ssz1 Y351Bpa crosslinking (Supplementary Fig. 3) is weak. The absence of fragments from the proposed Ssb1 peptide with an Arginine is surprising (S.F. 3B). The assignment of Ssb1 peptide was essentially only based on mass, which is not likely conclusive for trypsin and Asp-N double digestion. There was not a compelling separation between the candidates and decoys in StavroX decoy analysis. The proposed Ssb1 crosslinked peptide has a miscleavage site. Was the completely digested version of this crosslink pair detected? How about the version of peptide without Methionine oxidation? If there is no additional MS data to support this assignment, the authors should include relevant discussions and probably justify why Ssb1 R540 and R545 were chosen for later study presented in Fig. 2.

Response: Peptides containing both oxidized and non-oxidized versions of methionine were identified; the oxidized version has a higher Stavro scoring so it was listed preferentially. We have exchanged the data in Supplementary Figure 3 with that for the (⁶⁰⁹[AMSSR]⁶¹³) peptide (present in 4 of 5 dipeptides that were above the false discovery rate of 1%), rather than the incomplete digestion product for the Ssb1 peptide. The data show that β chain fragment ions are present but in lower abundance than α chain fragments. However, both b- and y- ions containing crosslinked α and β chains are present in high abundance. Note: the digest was trypsin alone, not trypsin and AspN as indicated in Table 1.

Regarding the choice of Ssb1 residues: Initially, positions in SBD α were chosen because the modeling suggested its interaction with Ssz1. We would have done the same in vivo analysis because of the docking model even if we had not had any mass spectrometric data from 351. 545 was continued throughout because it crosslinked to both Ssz1 and the ribosome directly and therefore gave us a direct relative comparison. 540 provided an additional SBD α position to assess Sse1 crosslinking, and thus also as one of the "tests" our model.

2. The comparison of Ssb1/Ssz1 and Ssb1/Sse1 interactions was confusing both in text and Fig. 2B. The position of Ssb1 NBD should be consistent in both images to illustrate the difference in ATP and ADP state, as well as interaction surfaces. Similarly, it will be visually helpful to align Ssb1 and Ssz1 as how they form the complex in Sup Fig. 4C for readers to appreciate the charge complementarity.

Response: We agree with the reviewer and have rewritten the portion of the text that deals with Sse. We also changed the orientation of the complex in Fig 2B as requested. In addition, as requested, we added the complex to Supplementary Fig 4C.

3. For MS data acquisition in Method section, resolution, mass isolation window and mass analyzer for MS1 and MS2 data should be specified and matched to the search parameters in Stavrox.

Response: We added the requested information to the Methods section.